# Cerebral Toxoplasmosis as an Uncommon Complication of Biologic Therapy for Rheumatoid Arthritis: Case Report and Review of the Literature

**DOI:** 10.3390/brainsci12081050

**Published:** 2022-08-08

**Authors:** Gonçalo Borges de Almeida, Miguel Cristóvão, Carlos Pontinha, Gonçalo Januário

**Affiliations:** 1Department of Neuroradiology, Hospital de São José, 1150-199 Lisbon, Portugal; 2Department of Pathology, Hospital de São José, 1150-199 Lisbon, Portugal; 3Department of Neurosurgery, Hospital de São José, 1150-199 Lisbon, Portugal

**Keywords:** cerebral toxoplasmosis, rheumatoid arthritis, biologic therapy, adalimumab

## Abstract

Toxoplasmosis is one of the most common opportunistic infections, mainly reported in patients with acquired immunodeficiency syndrome (AIDS). Patients with rheumatoid arthritis (RA) have also been linked to reactivation of toxoplasmosis due to immunosuppressive treatment, although biologic drugs have seldom been implicated. We present a case of cerebral toxoplasmosis in a 62-year-old female patient with RA after initiation of biologic therapy (adalimumab). The patient had detectable serum IgG antibodies to *toxoplasma gondii*, was also on chronic treatment with other non-biologic drugs and presented with worsening disorientation, unsteady gait and left hemiparesis. Imaging studies showed a space-occupying lesion in the right basal ganglia with ring-enhancement. Brain biopsy confirmed the diagnosis of toxoplasmosis and the patient was treated with pyrimethamine and sulfadiazine for 6 weeks, showing complete recovery on follow-up. A review of the literature yielded other four case reports of cerebral toxoplasmosis implying biologic drugs; however, data concerning toxoplasmosis serologic testing, prophylaxis and treatment in these patients are lacking. Each case must be carefully evaluated prior to treatment and a high-index of suspicion in seropositive patients is warranted. Since the use of biologic drugs is increasing, further research is needed to establish practical guidelines for seropositive patients receiving immunosuppressive treatment.

## 1. Introduction

Toxoplasmosis is one of the most common opportunistic infections and parasitic zoonoses worldwide. It is caused by the intracellular protozoan parasite *Toxoplasma gondii* (TG), usually hosted in cats and believed to infect about one third of the world’s population [1]. Human infection normally occurs following ingestion of water and food contaminated with cat feces or ingestion of toxoplasma cysts in undercooked meat or vegetables [2,3]. It may also be transmitted to the fetus when infection happens during pregnancy, resulting in congenital toxoplasmosis.

Three morphological forms are recognized in the life cycle of TG: the oocyst (containing sporozoites), tachyzoite and tissue cyst (containing bradyzoites). While the oocyst is found exclusively in the gut of cats, both the tachyzoite and bradyzoite are found reproducing and disseminating inside the human body. The tachyzoite represents a fast disseminating form of the parasite and the bradyzoite constitutes a slow dividing and dormant stage of TG. These bradyzoites are mainly located in the brain and muscle and held inside cysts, which helps them evade host immune responses and establish long-term persistent infection without symptomatic disease [4].

Primary infection in immunocompetent patients is usually asymptomatic or presents as a self-limiting illness. Clinical disease is usually seen following reactivation of latent TG infection in immunocompromised patients, in which dormant bradyzoites transform into rapidly dividing tachyzoites (phenotypic switching) [1,4]. In these cases, toxoplasmosis may cause life-threatening disease, especially with central nervous system (CNS) involvement [1].

CNS toxoplasmosis occurs in about 30–40% of TG-seropositive patients with human immunodeficiency virus (HIV) infection [5], usually with acquired immunodeficiency syndrome (AIDS). In addition, it has also been reported in other immunocompromised states such as chemotherapy, solid organ or stem cell transplants and autoimmune diseases. In fact, the pathogenesis of autoimmune diseases usually implies CNS infiltration by T-cells, which theoretically initiate an immune response against TG [3]. However, the majority of these patients are on chronic treatment with immunosuppressors and/or immunomodulators, which deplete T-cell infiltrates in the CNS and may increase the risk of developing opportunistic infections.

A systematic review of the literature revealed an interesting overall toxoplasmosis seroprevalence of 46% in patients with rheumatoid arthritis (RA), while the overall seroprevalence in non-RA patients is only 21% [6]. These findings suggest that toxoplasmosis may contribute to the pathogenesis of chronic inflammatory diseases as it triggers the immune system. The higher seroprevalence of toxoplasmosis in RA patients and the long-term use of immunosuppressive drugs in autoimmune disorders increase the likelihood of toxoplasmosis infection in RA patients. Interestingly, cerebral toxoplasmosis due to biologic therapy with tumor necrosis factor α (TNF-α) inhibitors in patients with RA has been scarcely reported in the literature [7].

The aim of this work is to present a case of cerebral toxoplasmosis in a patient with RA following initiation of adalimumab, a TNF-α inhibitor, and review the existing literature of similar cases due to biologic therapy for RA.

## 2. Materials and Methods

Clinical, imaging and pathologic data were retrospectively obtained from the patient’s case files and clinical notes. Written informed consent was obtained from the patient after discharge.

A review of the literature was conducted on PubMed using the research terms: “Brain toxoplasmosis” or “cerebral toxoplasmosis” AND “rheumatoid arthritis” AND “biologic therapy” or “TNF-α inhibitors”. All papers were screened by title, abstract and main text. We selected papers written in English and reporting at least one case of cerebral toxoplasmosis in patients with RA treated with biologic agents. For each case report/case series, the following information was obtained: RA drug treatment, toxoplasmosis serology, diagnostic methods, toxoplasmosis drug treatment and prognosis.

## 3. Results

A 62-year-old female with a known medical history of severe RA, non-insulin dependent diabetes mellitus and hypothyroidism presented to our hospital due to worsening temporal disorientation and unsteady gait over a two-week period. The patient had been diagnosed with RA 10 years earlier and was on chronic treatment with oral prednisolone (10 mg daily), leflunomide (20 mg daily) and methotrexate (7.5 mg once every week). Due to progressive joint pain and deformity, adalimumab (40 mg/0.4 mL once every two weeks) was added to her regimen 2 months earlier, while continuing to receive methotrexate, leflunomide and prednisolone.

Neurologic examination revealed moderate left hemiparesis (grade 3 in the medical research council [MRS] scale), with no evidence of spatial disorientation, sensory disturbances or cranial nerve injuries. A laboratorial work-up was carried out and showed a normal white blood cell count (9100 µL) with relative neutrophilia (8120 µL, 89%) and relative lymphopenia (600 µL, 7%), as well as and hyperglycemia (136 mg/dL) with high HbA1c (6.9%). The patient was seronegative for HIV type 1 and had detectable serum Immunoglobulin G (IgG) antibodies to TG, although Immunoglobulin M (IgM) titers were negative. The remaining analytical parameters, electrocardiogram and chest radiography did not show abnormalities and the patient was afebrile (T = 36.4 °C).

Computed tomography (CT) scan of the brain revealed the presence of a focal space-occupying hypodense lesion in the right basal ganglia, surrounded by edema. The lesion caused mass effect with effacement of ipsilateral cerebral sulci, buckling of the ventricular system (without hydrocephalus) and mild deviation of midline structures to the left (Figure 1). On magnetic resonance imaging (MRI), the lesion was relatively heterogeneous on T2/FLAIR and exhibited marginal enhancement following gadolinium injection. Perfusion-weighted imaging demonstrated significant decrease of cerebral blood flow (CBV) within the lesion (Figure 2). Following imaging characterization of the lesion, an intraoperative consultation with brain biopsy was performed through a right parietal burr hole. A brain smear revealed the presence of brain tissue with abundant dirty necrosis and predominantly acute inflammatory cells. Cystic structures containing encapsulated microorganisms were also identified, raising the hypothesis of TG infection. A brain biopsy was later performed and revealed not only cysts containing bradyzoites but also free microorganisms (tachyzoites) across the sample. Immunohistochemical staining with antibodies against TG confirmed the diagnosis of cerebral toxoplasmosis, highlighting the free tachyzoites (Figure 3). A post-biopsy CT scan excluded procedure-related complications.

The patient did not own a cat and no other common risk factors for toxoplasmosis were identified. We hypothesize that the transmission route was probably related to ingestion of contaminated food or water in the past.

The patient was given dexamethasone 0.5 mg daily to decrease brain edema and started anti-toxoplasma treatment with pyrimethamine (PYR, loading dose of 100 mg PO followed by 75 mg/day), folinic acid (10 mg/day) and sulfadiazine (SDZ, 1 g PO q6h). In addition, adalimumab was discontinued. The patient showed mild clinical and radiological response during the first 3 weeks. Since she did not experience severe joint pain in the previous weeks, prednisolone was gradually reduced to 5 mg. Following de-escalation of prednisolone, the patient showed better clinical response with progressive motor improvement, presenting mild hemiparesis at discharge (grade 4 in the MRS scale). She completed a six-week total course of PYR-SDZ. Before discharge, blood tests were drawn and showed resolution of the relative neutrophilia (2980 µL, 49%) and relative lymphopenia (1950 µL, 32%).

The patient was kept on follow-up appointments after discharge and fully recovered from the motor deficit after a six-month follow-up period. She received treatment with oral prednisolone (5 mg daily), leflunomide (20 mg daily) and methotrexate (7.5 mg once every week) and did not experienced severe joint pain after discharge. In addition, she received prophylactic treatment with trimethoprim-sulfamethoxazole (TMP-SMZ, 160/800 mg PO twice daily) due to the risk of toxoplasma reactivation.

## 4. Discussions

Toxoplasmosis is one of the most common opportunistic infections, mainly reported in HIV-positive patients and following solid organ and/or stem cell transplantation. Data concerning the incidence and prevalence of toxoplasmosis in HIV-negative, non-transplant immunocompromised patients are scarce. In addition, the optimal treatment and prophylaxis in patients taking immunosuppressive drugs have yet to be determined, with no clear strategy to identify high-risk patients, choose the best prophylactic treatment and establish the optimal length of treatment course [3]. The small number of case reports and lack of randomized controlled trials provide little evidence for toxoplasmosis prophylaxis and treatment in patients receiving biologic therapy.

Despite the high seroprevalence of toxoplasmosis in patients with RA and the long-term use of immunosuppressive drugs in these patients, this infectious disease has seldom been reported as a complication of immunosuppressive treatment in patients with RA. The immunosuppressive treatment of RA consists of methotrexate, glucocorticoids, non-biological disease-modifying antirheumatic drugs (DMARDs), biologic agents or a combination of these drugs. Biologic therapy comprises TNF-α inhibitors (including infliximab, adalimumab, etanercept, certolizumab and golimumab), as well as other monoclonal antibodies (such as rituximab, anakinra, abatacept, ustekinumab, tocilizumab, belimumab and tofacitinib) [3].

Biologic agents are potent immunosuppressors which are known to increase the risk of opportunistic infections. In fact, a large-cohort study involving mainly patients with RA [8] showed an increased rate of non-viral opportunistic infections in patients treated with TNF-α inhibitors compared to those initiating non-biological DMARDs (adjusted hazard ratio 1.6; 95% CI 1.0 to 2.6). The difference was even larger concerning new users of infliximab vs. patients starting non-biological DMARDS (adjusted hazard ratio 2.6; 95% CI 1.2 to 5.6), Moreover, baseline corticosteroid use was also associated with an increased risk of non-vital opportunistic infections. Pneumocystosis, nocardiosis/actinomycosis and tuberculosis were the most commonly reported opportunistic infections, while toxoplasmosis was only seen in 1 out of 80 cases of non-viral infection [8]. Notably, the risk of infection seems to be higher among patients with long-term RA and shortly after initiation of biologic agents [9]. Concerning cerebral toxoplasmosis, it is believed that immunosuppression may allow phenotypic switching with rapid proliferation of tachyzoites, which may surpass the host’s already compromised immune response and cause life-threatening disease [4].

A review of the literature yielded another four case reports of cerebral toxoplasmosis in patients with RA treated with biologic therapy, all of which occurred in HIV-negative patients (Table 1). Only one case occurred due to monotherapy with a biologic agent [10], while the remaining three involved a combination of different drugs. Although the simultaneous use of more than one drug hinders the possibility of determining the causative drug(s), one of the case reports [11] showed a clear temporal relationship between the increase in the dose of infliximab and reactivation of toxoplasmosis. Similarly, our case report depicts reactivation of toxoplasmosis after initiation of adalimumab in a patient with RA already on chronic therapy with other non-biologic drugs. Therefore, we believe that TG reactivation resulted from a combination of immunosuppressive drugs, powered by the introduction of a biologic agent.

The diagnosis of toxoplasmosis implies polymerase chain reaction (PCR) of the cerebrospinal fluid (CSF) or brain tissue, serology or biopsy. Concerning serologic tests, it is not always easy to distinguish latent infection from reactivation of latent infection as both may present with positive IgG and negative IgM. In these cases, serological control should be obtained after 3 weeks: stable IgG levels indicate latent infection and increasing levels imply determination of IgG avidity. If IgG avidity is intermediate or low acute infection cannot be excluded, high IgG avidity strongly suggests reactivation [12]. Unfortunately, determination of IgG avidity is not routinely carried out at our hospital and only simple serologic testing was performed after adalimumab was introduced. Among the aforementioned case reports with available data on toxoplasmosis serology, none of them showed IgM positivity.

Regarding PCR as diagnostic method, sensitivity values for detection of TG in CSF have been reported to vary widely in the literature (44.4–100%), although specificity is usually very high [4,13]. Furthermore, a positive result of brain PCR tissue may not differentiate toxoplasma encephalitis from latent infection [14]. Finally, data concerning histopathology sensitivity and specificity for the diagnosis of cerebral toxoplasmosis in humans are scarce. The reason why we chose this diagnostic method is because brain biopsy is a frequently performed procedure at our hospital in the presence of a surgically accessible space-occupying brain lesion. Moreover, repeating TG serologic testing or performing PCR of CSF would require a high-index of suspicion for this uncommon complication of biologic therapy in a patient with relative neutrophilia and leukopenia but normal total white blood cell count. Similarly, all the other case reports carried out brain biopsy to establish the diagnosis, one of which was also confirmed by PCR of brain tissue [10].

The most effective treatment for cerebral toxoplasmosis is PYR-SDZ plus folinic acid (leucovorin) to prevent hematological complications associated with pyrimethamine [15]. In case of sulfa allergy, PYR and clindamycin may be an effective combination. Trimethoprim-sulfamethoxazole (cotrimoxazole) is mainly used for prophylaxis, although it may be used for toxoplasmosis treatment when PYR is not available. We treated our patient with PYR-SDZ but felt the need to adjust the immunosuppressive treatment to achieve better results. In fact, two of the other four case reports also discontinued immunosuppressive drugs: in one case, the patient showed improvement on follow-up [10] and in the other, the prognosis is unknown [7], as the patient was lost to follow-up after being transferred to a long-term care hospital. Concerning prophylaxis, our patient received TMP-SMZ after toxoplasmosis treatment to decrease the risk of reactivation. She did not receive prophylaxis prior to admission since her TG serologic status was unknown and we did not identify common risk factors for TG infection.

**Table 1 brainsci-12-01050-t001:** Summary of reported cases of cerebral toxoplasmosis in patients with RA treated with biologic drugs.

Reference	RA Drug Treatment	Toxoplasmosis Serology	Methods for Toxoplasmosis Diagnosis	Toxoplasmosis Drug Treatment	Prognosis
Young et al. [11], 2005	Prednisolone, methotrexate, leflunomide, **infliximab**	IgM: negative IgG: positive	Brain biopsy	PYR, folinic acid and dapsone (previous sulfa allergy)	Improved
Nardone et al. [10], 2014	**Adalimumab**	NR	Brain biopsy, PCR of brain tissue	PYR-SDZ and folinic acid; discontinuation of adalimumab	Improved
Pulivarthi et al. [16], 2015	Methotrexate, **infliximab**	IgM: negative IgG: positive	Brain biopsy	PYR, leucovorin and clindamycin (previous sulfa allergy)	Neurologically stable
Hill et al. [7], 2020	Methotrexate, **infliximab**	IgM: unknown IgG: positive	Brain biopsy	TMP-SMZ; discontinuation of all RA drugs	NR
Our case report	Prednisolone, methotrexate, leflunomide, **adalimumab**	IgM: negative IgG: positive	Brain biopsy	PYR-SDZ and folinic acid; discontinuation of adalimumab	Full recovery on follow-up

*IgG*, Immunoglobulin G; *IgM*, Immunoglobulin M; *NR*, Not reported; *PCR*, Polymerase chain reaction; *PYR*, Pyrimethamine; *RA*, Rheumatoid arthritis; *SDZ*, Sulfadiazine; *SMZ*, Sulfamethoxazole; *TMP*, Trimethoprim. TNF-α inhibitors (biologic agents) are shown in bold.

The differential diagnosis of cerebral toxoplasmosis is extensive and includes lymphoma, metastasis and other opportunistic infections, as these entities may share similar imaging features [2]. Nonetheless, the presence of a small mural nodule within the ring-enhancing lesion (“eccentric target sign”) and/or the existence of concentric alternating zones of hypo and hyperintensity on MRI (“concentric target sign”) suggest the diagnosis of toxoplasmosis [16]. Moreover, low metabolic activity on thallium single-photon emission computed tomography (SPECT) and/or positron emission tomography (PET) and decreased CBV on perfusion-weighted MRI favor the hypothesis of a pseudotumoral mass (such as infection) instead of a brain tumor or metastasis [2].

## 5. Conclusions

This case report portrays reactivation of toxoplasmosis after initiation of adalimumab in a patient with RA already on chronic treatment with other non-biologic immunosuppressive drugs. Although there is no current evidence for serologic testing for toxoplasmosis prior to treatment or primary prophylaxis if positive, these patients should be carefully evaluated on a case-by-case approach to weigh the risks and benefits of immunosuppressive treatment. A high-index of suspicion in seropositive patients is warranted to allow prompt treatment, if necessary. Since the use of biologic drugs in chronic inflammatory diseases is increasing, further research is needed to establish practical guidelines concerning toxoplasmosis serologic testing, prophylaxis and treatment in patients receiving immunosuppressive treatment. These patients should remain on immunomodulation follow-up appointments to provide better management of these complex clinical contexts.

## Figures and Tables

**Figure 1 brainsci-12-01050-f001:**
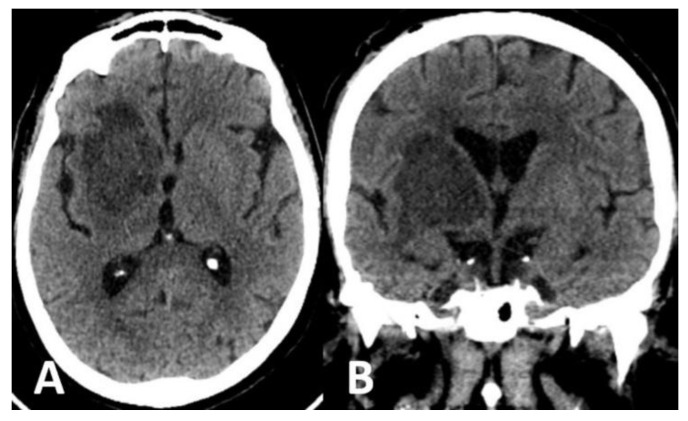
CT scan of the brain (**A**,**B**) showed a space-occupying hypodense lesion in the right basal ganglia, surrounded by edema involving the frontal and temporal lobes. Mass effect was seen with effacement of cerebral sulci, compression of the ventricular system and mild left deviation of midline structures.

**Figure 2 brainsci-12-01050-f002:**
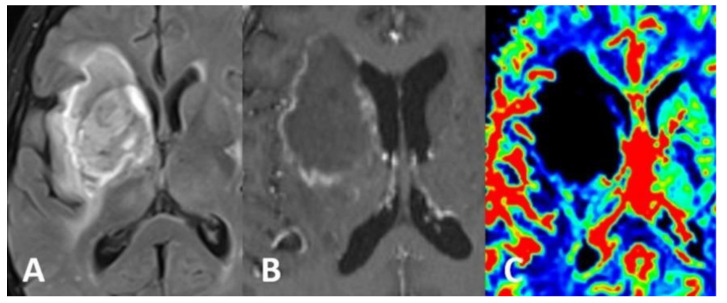
MRI of the brain showed a heterogeneous lesion on the T2/FLAIR sequence (**A**), displaying ring-enhancement after gadolinium injection (**B**). Perfusion-weighted imaging showed significant decrease of CBV (**C**) within the lesion.

**Figure 3 brainsci-12-01050-f003:**
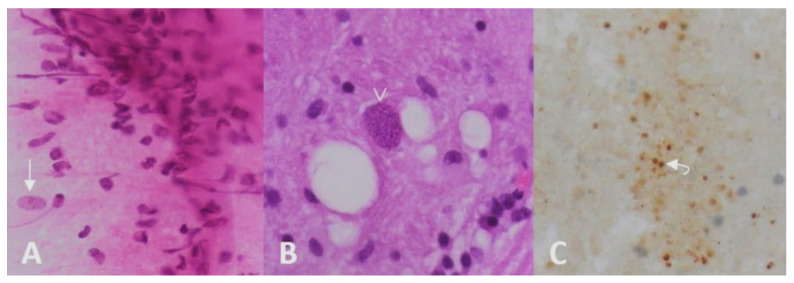
Brain intraoperative smear (**A**) showed cysts with bradyzoites (*arrow*) in a background of glial tissue. Definitive histopathological exam (**B**) confirmed the presence of bradyzoite cysts (*arrowhead*) and chronic inflammatory cells. Immunohistochemical stain (**C**) with anti-Toxoplasma antibody highlighted the more elusive free tachyzoites (*curved arrow*).

## Data Availability

Data are available from the corresponding author upon reasonable request.

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
