# Peer review of "Cerebral Toxoplasmosis as an Uncommon Complication of Biologic Therapy for Rheumatoid Arthritis: Case Report and Review of the Literature"

_brainsci, 2022, doi:10.3390/brainsci12081050_

Round 1

Reviewer 1 Report

 The authors described a 62-year old female with RA, who were complicated with cerebral toxoplasmosis after initiation of adalimumab. Similar case reports have been already published but this severe complication is not widely recognized and is valuable to report. Major comments 1. What was the transmission route? Did she have cats? 2. Table indicated that adalimumab was used with MTX, leflunomide and glucocorticoid. But the Result section, adalimumab was added while continuing to receive MTX. Probably, glucocorticoid (PSL10mg) also continued. Oral PSL 10mg is high dose for RA and can be an important risk factor for toxoplasma infection, which is described in Discussion section. Combination therapy of immunosuppressive drugs seems more important risk for cerebral toxoplasmosis in this case. 3. After cerebral toxoplasmosis, how is her RA treated? Does she receive prevention therapy for TG? Please describe in the case presentation section the patient was lost to follow-up after being transferred to a long-term care hospital. 4. Patient consent form is needed and please mention it in methods section.

Reviewer 2 Report

The authors presented a case of cerebral toxoplasmosis in a 62-year-old female patient with RA after starting biologics therapy (adalimumab). This case is one in a row of RA patients receiving biologic therapy, with additional information on serologic toxoplasmosis testing, prophylaxis, and treatment that is lacking in other cases. The manuscript overall is very well written, and I am glad that the authors decided to include more about toxoplasmosis testing, prophylaxis, and treatment.

So, to make these three differences more prominent, some changes should be made.
My questions and suggestions are:
In their manuscript, the authors listed prophylaxis as one of the main differences from other similar cases. My question is: Did your 62-year-old patient take prophylaxis? If not, this should be mentioned.

Also, in line 203, next to the sentence "In addition, the optimal treatment and prophylaxis in patients taking immunosuppressive drugs have yet to be determined, with no clear strategy to identify high-risk patients, choose the best prophylactic treatment, and establish the optimal length of treatment course[3]." ....shortly explain why is the prophylaxis so difficult to implement before biologic therapy? (for a wider readership)

The authors explained serologic testing in the case of TG(as one of the main differences from other similar cases):
1. was serology performed (IgG and IgM) on TG before adalimumab was added to the therapeutic regimen? If not, this should be mentioned in the discussion part.
2. also, the authors mentioned IgG testing after 3 weeks and an avidity test as well, to be sure and to rule out acute infection with TG. Is this testing a part of the usual protocol at your hospital or not, before adalimumab was added? If not, this should be mentioned next to the explanation about serology in the discussion part (line 288).

The authors mentioned that the diagnosis of toxoplasmosis (as one of the main differences from other similar cases) requires polymerase chain reaction (PCR) of the CSF, serology, or biopsy (line 280).
It is clear from the explanation that serology and PCR of CSF were not the main focus, but brain biopsy was. Please include an explanation (to make it clearer for anyone reading the text) why brain biopsy was the first line of diagnostic testing in your case.

The authors explained that serology is not sensitive enough because IgM is often negative in such reactivated TG. However, explain why the team did not wait 2 weeks for a second blood taking to measure IgG again or performed the PCR of CSF first. As a diagnostic method, a CNS biopsy is more invasive and associated with more complications than a blood and CSF taking. So why not wait? This should be explained in more detail for a wider readership.

Round 2

Reviewer 1 Report

The manuscript has been revised well.

Reviewer 2 Report

The authors improved their manuscript.